# Manifestations, responses, and consequences of mistreatment of sick newborns and young infants and their parents in health facilities in Kenya

**Timothy Abuya**[1]*, **Charlotte E. Warren**[2], **Charity Ndwiga**[1], **Chantalle Okondo**[1], **Emma Sacks**[3], **Pooja Sripad**[2]

**1** Population Council, Nairobi, Kenya, **2** Population Council, Washington, DC, United States of America, **3** Johns Hopkins School of Public Health, Baltimore, MD, United States of America

* tabuya@popcouncil.org

## Abstract

**Data Availability Statement:** There are institutional legal restrictions on sharing de-identified data sets, however, anonymized / aggregated datasets may

### Background

Despite efforts to incorporate experience of care for women and newborns in global quality standards, there are limited efforts to understand experience of care for sick newborns and young infants. This paper describes the manifestations, responses, and consequences of mistreatment of sick young infants (SYIs), drivers, and parental responses in hospital settings in Kenya.

### Methods

A qualitative formative study to inform the development of strategies for promoting family engagement and respectful care of SYI was conducted in five facilities in Kenya. Data were collected from in-depth interviews with providers and policy makers (n = 35) and parents (n = 25), focus group discussions with women and men (n = 12 groups), and ethnographic observations in each hospital (n = 64 observation sessions). Transcribed data were organized using Nvivo 12 software and analyzed thematically.

### Results

We identified 5 categories of mistreatment: 1) health system conditions and constraints, including a) failure to meet professional standards, b) delayed provision of care; and c) limited provider skills; 2) stigma and discrimination, due to provider perception of personal hygiene or medical condition, and patient feelings of abandonment; 3) physically inappropriate care, including providers taking blood samples and inserting intravenous lines and nasogastric tubes in a rough manner; or parents being pressured to forcefully feed infants or share unsterile feeding cups to avoid providers' anger; 4) poor parental-provider rapport, expressed as ineffective communication, verbal abuse, perceived disinterest, and non-consented care; and 5) no organized form of bereavement and posthumous care in the case of infant's death. Parental responses to mistreatment were acquiescent or non-confrontational

be available from the Population Council data repository team upon reasonable request. Request may be send to Population Council, Dataverse using the email; publications@popcouncil.org.

**Funding:** Population Council implemented the study with funding from the United States Agency for International Development (USAID) under Breakthrough RESEARCH (Cooperative Agreement no. AID-OAA-A-17-00018). There were no other sources of funding. The funders had no role in study design, data collection and analysis, decision to publish, or preparation of the manuscript. The authors received a salary from other projects in addition to the current projects. Timothy Abuya, Charity Ndwiga and Chantalle Okondo received salaries from another USAID project funded under the federal number: 7200AA19CA00026.- named Kuboresha Afya Mitaani: Urban MNCH Project. Charlotte Warren and Pooja received salary from another Bill & Melinda Gates Foundation project under agreement number- ID OPP1174594. Emma Sacks received salary from Johns Hopkins School of Public Health.

**Competing interests:** There are no competing interests.

**Abbreviations:** SYI, sick young infant (0–59 days).

and included feeling humiliated or accepting the situation. Assertive responses were rare but included articulating disappointment by expressing anger, and/or deciding to seek care elsewhere.

## Conclusion

Mistreatment for SYIs is linked to poor quality of care. To address mistreatment in SYI, interventions that focus on building better communication, responding to the developmental needs of infants and emotional needs for parents, strengthen providers competencies in newborn care, as well as a supportive, enabling environments, will lead to more respectful quality care for newborns and young infants.

## Introduction

In the last decade, mistreatment of women giving birth in facilities has received global attention [1–5]. Negative experiences during labor and childbirth are a deterrent to the choice of birth in health facilities [3, 6, 7], which has implications for obstetric, neonatal, and pediatric health. While there has been an exponential growth in the number of studies examining mistreatment of women during childbirth, few studies have looked at the experience of women, their partners, and their very young children when receiving or seeking care for their sick newborns or young infants. Several publications have proposed definitions and conceptual frameworks for understanding mistreatment [1, 3, 8, 9] and covered a spectrum of unpleasant to traumatic experiences, many of which are violations of women's rights [10] and are limited to mostly maternity care. Bowser and Hill pioneered an approach to defining "disrespect and abuse" for maternity care [9], which subsequently led to a typology with seven overarching categories of mistreatment including verbal abuse, sexual abuse, physical abuse, stigma and discrimination, failure to meet professional standards, poor rapport between women and providers and health system conditions and constraints [3].

Attention to the experience of care of the newborn is more recent. The Respectful Maternity Care (RMC) Charter, published in 2011 and updated in 2019, articulates 10 fundamental rights of childbearing women and newborns. Inclusion of the newborn was essential to provide a framework for understanding the combined mother and newborn experience of care [11]. The World Health Organization (WHO) released a statement on the elimination of disrespect and abuse of women in childbirth in 2014 [7] and subsequently a framework for quality of care for maternal and newborn health (MNH) in 2016 [12]. The WHO MNH quality of care framework comprises eight domains including both the provision of care for and the experience of care by women and newborns in health facilities. Experience of care consists of effective provider communication with women and their families about the care provided, parents' expectations and rights, care with respect and preservation of dignity, and access to social and emotional support for care received or events that may present during care. The cross-cutting areas of both experience and provision of care include availability of competent, motivated human resources and the physical resources that are prerequisites for good quality [12]. More recently, WHO published standards for improving the quality of care for small and sick newborns in health facilities, which articulates the concept of experience of care from the newborn perspective with three standards focusing on the needs and rights of newborns and their families along the care process [13].

Despite attempts to incorporate experience of care for women and newborns into global quality standards [13], there have been limited efforts to understand experience of care for sick newborns and young infants and their families seeking routine services or emergency care for critical illness [14, 15]. The complexity of conceptualizing experience of care by newborns and young infants is compounded by their inability to verbally express their needs or share their experiences [15]. Newborns and young infants express their experience of care with signs of distress or calm depending on the sensorial environment, indicating the importance of the caring environment and how this can impact on their development. Due to the long-lasting impact on the infant, there is need for providing respectful and nurturing care that responds to the needs of the infant [16]. This paper explores the experience of care of parents—mothers and fathers (or other close relatives)—who sought care for their sick young infants (SYIs; aged 0–59 days) in five large hospitals in Kenya. Drawing on perspectives of parents and providers, we describe what constitutes mistreatment of SYIs, what drives provider behaviors, parents' immediate responses to mistreatment of SYIs, and the consequences of these experiences on parents.

## Materials and methods

### Design and participants

This was a qualitative formative study designed as part of larger implementation research to develop and test strategies for improving provision of nurturing care; promotion of family engagement; and communication and respect for care of newborns, infants, and very young children in resource-constrained facilities in Kenya. This paper describes the experience of care for parents and their SYIs (0–59 days) from the perspective of parents and providers.

### Study sites

The study was conducted in five facilities in Nairobi and Bungoma counties, which are urban and rural, respectively. Both counties have neonatal mortality rates higher than the national average of 22 /1000 live births: 39/1000 in Nairobi and 33/1000 in Bungoma. Infant mortality rates in the counties are also higher than the national estimates of 39/1000 live births, with Nairobi estimating infant mortality at 60/1000 live births and Bungoma at 97/1000 live births [17]. Study facilities were selected in consultation with the county health management teams and the national Ministry of Health. In Nairobi, data were collected from three facilities: one large public maternity hospital, one large tertiary public hospital, and one faith-based hospital. In Bungoma, data were collected from two public referral hospitals.

### Data sources

**In-depth interviews with policy makers and healthcare providers.** In-depth interviews (IDIs) were conducted with health officials at the national, county, and sub-county level and with providers working in newborn and pediatric units in the selected facilities. The interviews focused on understanding the policy contexts that support or hinder provision of respectful and nurturing care, experience of care for SYIs, and interactions between SYIs' parents and their health providers.

**Joint in-depth interviews with parents.** We recruited parents of SYIs receiving care in the five hospitals and invited both parents of the SYI to participate in a joint interview. Efforts were made to include parents whose infant had died to explore the specific experience of bereavement care. Interview topics included experiences of hospitalization, interactions with providers and the health system, and experience of respectful care for SYIs. In addition, we

explored parents' perceptions on partnering with providers to improve care and management of their SYI, including emotional and social support. Interviews were conducted by research assistants with training in qualitative data collection using an IDI guide. Interviews were conducted in Kiswahili or English, depending on the participants' fluency.

**Focus group discussions.** In each of the two counties, we conducted separate focus group discussions (FGDs) with women and men with SYIs who were hospitalized or had been born small or sick (Table 1). All the focus group participants were purposively identified through study facilities. The FGDs focused on understanding the normative perspectives of families managing SYIs, including normalized mistreatment and provider-parent interactions, and the typical parental response in hospital settings. Each FGD was facilitated by two trained research assistants, conducted in Kiswahili, and held in locations that were convenient for participants. Sessions were audio-recorded with the consent of participants. When necessary, FGDs were paused to allow time for breastfeeding or urgent attention to SYIs. In cases of severe anxiety and emotional stress of parents who had experienced mistreatment or an infant death, efforts were made to link them with professional support within the facility.

**Ethnographic observations.** We conducted non-participatory observations of the hospital settings which helped us explore routine care processes, infrastructure, and the environment in which coordination of services and interactions between providers and parents take place. For each study facility, observations were conducted for ten consecutive days, including weekends, in various departments where SYIs are managed. We recorded information using a check list template that captured how providers plan daily schedules to provide care for infants and execute their work, and how providers interact with parents and families including during bereavement or critical moments such as when the child required emergency attention. We also tracked the process of care from registration to discharge, documenting all service points and care provided in each location. A pair of research assistants, one with a clinical background and another with ethnography skills, documented the dynamic care processes for services provided. This helped to understand the experiences of mistreatment and opportunities for the health system to provide a better partnership between providers and parents of SYIs. Details of the number of participations interviewed are presented in Table 1.

**Data processing and analysis.** Each day, the field team documented their daily reflections, including practical challenges such as working with facility administrators to ensure proper privacy protection for interviewees, meeting recruitment goals for male participants, or finding space for large FGDs close to the locations of the SYI. The field reflections and observational data were submitted to a centralized location for synthesis and storage. Qualitative data from the IDIs and FGDs were transcribed verbatim and Kiswahili and other local language transcripts were translated into English.

Thematic analysis, adapting existing mistreatment frameworks [3, 15], guided our deductive and inductive analytical approach to coding and interpreting textual data from the

**Table 1. Summary of study activities and number of participants.**

| Methods | Nairobi | Bungoma | Total |
|---|---|---|---|
| IDIs with policy makers | 2 | 1 | 3 |
| IDIs with providers and managers | 18 | 14 | 32 |
| IDIs with both parents | 11 | 12 | 23 |
| IDIs with single parents | 1 | 1 | 2 |
| FGDs with mothers | 5 | 3 | 8 |
| FGDs with male partners | 2 | 2 | 4 |
| Ethnographic observations | 33 | 31 | 64 |

transcripts. Our deductive approach entailed classifying types of mistreatment that parents of SYIs experienced and understood or observed happening to SYIs. At the interpretive stage we used categories within Bohren et al's typology in maternal health [3] and Sacks' additional classifications around newborn experience [15]. This approach was complemented and preceded by an inductive approach to reviewing transcripts and listing preliminary themes arising from the data. A team of 3 researchers reviewed transcripts, field notes and observations, making initial annotations around topics arising from the data. An initial codebook of 19 themes was refined based on the process of open coding and progressive categorization of issues that emerged. Transcripts were exported into Nvivo 12 (QSR International) software with facility names redacted and coded by 3 researchers. This was followed by a process of summarizing mistreatment types experienced by SYIs and their parent's responses to this negative experiences. This process enabled the research team to explore similarities and differences in mistreatment and responses across sites and by SYI age category (between 0–59 days). The deliberative process looking at summaries in relation to the extant literature led to themes aggregated in the deductive categories that aligned closely with existing frameworks used to describe mistreatment of women during childbirth and the subsequent responses [3, 15, 18]. We adapted these categories to the SYI population during the interpretation and presentation of our results. For example, we used responses to mistreatment described by McMahon [18] and expanded it to extrapolate their effects categorized as *short-term* and *long-term consequences*. The responses were either *acquiescent measures*, which are non-confrontational methods, or *assertive measures*, which are more confrontational methods to address mistreatment.

**Ethical issues.** Participants were asked sensitive questions about the treatment of SYIs in hospitalized settings. To avoid the risk of others overhearing informants' information, interviews were conducted in comfortable private locations, with ample time for data collection to guarantee privacy and confidentiality. Researchers were trained to ensure that guidance on ethical conduct was clearly understood and implemented. The research team was trained to listen and observe intently without displaying any judgmental attitude. Study information was read to potential participants, and once they understood and accepted, participants signed the informed consent form. Informed consent forms and questionnaires for parent interviews and FGDs were translated into Kiswahili. The research protocol was approved by the Population Council's Institutional Review Board (PC IRB 893) and AMREF Ethical Review Board ESRC P646/2019 based in Kenya.

## Results

We provide details of mistreatment integrated with the drivers, immediate responses, and their effects. Reported examples of mistreatment fall into five third order categories [3] identified previously in the literature and a combined category identified as physically inappropriate care which includes inappropriate feeding practices and various forms of physical abuse. We operationalize the categories based on various manifestations and the underlying drivers of mistreatment as they were either observed or reported. Details of the categories of mistreatment are presented in Table 2.

### Manifestations and drivers of mistreatment of sick young infants

**a) Failure to meet professional standards and health system conditions and constraints.** In adapting the Sacks framework, we distinguish between failure to meet professional standards by providers and the health system ability to provide adequate standards of care. The former is related to staff knowledge, attitudes, and behaviors while the latter is related to the health system constraints which then drive mistreatment. The complexity of this

**Table 2. Categories of mistreatment experienced by newborns and reported drivers.**

| Third order themes | Second order themes | Illustrative manifestations of experiences and reports of first order themes |
|---|---|---|
| **Failure to meet professional standards and Health system conditions and constraints** | Abandonment & Neglect | Busy providers with too much workload to provide necessary care in a timely manner |
| | | Failure to monitor treatment procedures on infants, e.g. infants detach the IV 'tubes' or they get blocked leading to treatment interruption |
| | Delayed provision of care | Drugs and supplies shortages in facilities delay initiation of treatment and laboratory investigations |
| | | Provider laxity or lack capacity leading to infants having to wait for long periods to get treatment |
| | | Lack of accountability leads to some level of carelessness, like losing files amongst important documents, or not taking all the required tests leading to delays in initiating care for the infants |
| | | Poor facility readiness to accommodate SYIs who have been referred from other facilities; delays initiation of care |
| | Crowded conditions | Few cots leading to 4–5 SYIs sleeping together in one cot |
| | | Crowded emergency rooms compounded by slow triaging, delaying services |
| | | Insufficient incubators: infants put together detach each other's tubes, interrupting treatment |
| | | Sharing of beds by mothers and their newborns in postnatal ward |
| | Poor provider skills | Providers with limited neonatal skills give high dose of medication to infants or give wrong diagnosis |
| | Insufficient equipment | Use of wrong equipment due to lack of supplies and equipment, e.g. wrong size Ambu bag |
| | Harsh environmental conditions | Unnecessary exposure to cold as parents instructed to undress infants for weighing before their turn, exposing them to cold or infant being cleaned with cold tap water as hot water not available |
| | Poor hygiene practices | Providers fail to wash or sanitize in between handling infants or conduct procedures or instances where weighing scale used is soiled |
| | Unclear care processes | Lengthy discharge process where providers prioritize sick infants due to workload, leading to delays in discharging stable children |
| | Non-consented care | Religious beliefs that prohibit infants from being transfused blood or injected makes providers initiate care without consent to avoid delays or worsening of conditions |
| | | Trainees end up performing more roles than stipulated ones without close supervision, some provide services without parental consent |
| | | Blood samples and other tests done without parental consent |
| **Stigma and discrimination** | Discrimination due to socioeconomic status and poor personal hygiene | Mothers deemed "dirty" ordered to use separate incubators or instructed to weigh their infants last |
| | | Providers relate well with infants with disposable diapers compared to those with toweling or other cloth nappies |
| | Medical discrimination | Parents with HIV-exposed infants were attended to in a segregated area with screens |
| **Physically inappropriate practices** | Use of force | Perceived hard slapping of the newborn to cry soon after delivery |
| | Exposure to pain | Unnecessary pricking of infants before inserting an IV line |
| | Rough handling | Rough insertion and removal of oxygen tubes |
| | | Providers forcefully examining infants when they are not calm |
| | | Rough handling of newborns leading to injuries and fractures |
| | Unsuitable cots for infants | Failing to secure the cots leading to infants falling to the ground |
| | Forceful feeding | Parents forcefully feeding infants to avoid being scolded by providers |
| | | Some in Kangaroo Mother Care unit forcefully feed newborns to try to increase weight gain for early discharge |
| | Insufficient feeding | Insufficient feeding for infants as mothers with multiple children find it difficult to feed more than one child in the given feeding time |
| | | Missed feeding opportunities as infants are accidentally skipped by providers due to workload |
| | Poor hygiene feeding practices | Compromised hygiene as feeding cups are shared and may not be properly cleaned in the rush to feed |
| **Poor rapport between providers and parents and their infants** | **Verbal abuse** Harsh language or tone | Parents do not feel able to ask questions |
| Harassment especially when providers are asked questions, they respond harshly to the mothers. | | |

(*Continued*)

**Table 2.** (Continued)

| Third order themes | Second order themes | Illustrative manifestations of experiences and reports of first order themes |
|---|---|---|
| | Ineffective communication | Parents not given information on cord care, danger signs, identifying pain in children or what to do when pain persists, medication they are receiving and how it would help them |
| | | No clear communication on discharge process and on attachment of newborn to breast and general care of infant |
| | | Inadequate information regarding sample collection and test results for their infants, progress and follow up on the next visit |
| | Loss of autonomy | Parents are not consulted in the care of their child and feel unable to ask questions as providers are hostile |
| | | Inability to afford health care services or medication, leads to delayed discharge due to inability to clear bills |
| Bereavement and posthumous care | Lack of emotional support or counselling | No one counsels bereaved mothers. Parents not told why the infant died |

interaction illustrates the challenges of describing discrete categories of manifestations and what drives them. For example, both parents and providers reported examples of failure to meet professional standards during care of SYIs. Observations and parents' reports indicated that at times, providers took many breaks contributing to long queues of patients waiting to be seen. Another example of failure to meet professional standards by providers was reports of abandonment and neglect manifested through providers' failure to monitor treatment procedures or provide timely care because of poor organization of care, contributing to delays in care provision. These were also cited as driven by heavy workload of providers—a wider health system constraint:

> "...but once in a while, because sometimes we are busy, we don't have time to carry those babies, because you are the one preparing the milk, as well as clean them..." (IDI, Provider)

> "We waited and waited until later when we were told to go down there and start receiving services from there. We went there and the queue was long then we were again referred to come back upstairs, and we are yet to see the doctor" (FGD Women)

Delayed provision of care was also attributed to a lack of supplies and equipment, inadequate personnel to manage SYIs, and lack of accountability. For example, lack of facility readiness to accommodate families referred from elsewhere without prior communication delayed preparation of admission for such SYIs, postponing initiation of their care. There were also incidences where some level of carelessness, such as losing files or samples, forgetting results submitted for laboratory investigations, or not taking all the required tests led to redoing the samples, and contributed to delays in initiation of treatment for the SYI.

Some providers also expressed having limited skills to manage SYIs, especially treating those who presented with several common symptoms such as fever, cough, diarrhea. This contributed to delays in care due to inability of providers to insert intravenous lines. In an environment with limited diagnostic services, this presented a dilemma for providers who attempted to manage conditions but may have been perceived by caregivers to lack expertise to diagnose properly:

> "Once in a while you might make the wrong diagnosis not because you are not thorough but sometimes there are conditions that mimic others, you get a condition that you are suspecting based on history taken, you can suspect malaria and it wasn't." (IDI, Provider)

There are also instances of inaccurate drug administration:

"...at night ... I found my baby being administered paracetamol 5ml and he was two weeks and some days, I even asked is the medicine too much for the baby? And she was like "ooh sorry I forgot!" ...I even started doubting all the medication previously given to the baby...The whole thing just shocked me. The nurse was like "I had forgotten" so she took the medicine and reduced it to the right prescription." (FGD, Women)

Such practices were attributed to inadequate skills on the part of providers, inexperienced providers, limited opportunities for training in neonatal care or unavailability of guidelines and treatment protocols to facilitate care.

"On availability we have a challenge because guidelines are not things that you can put under key and lock. At times, when you bring them into provider room, they are mishandled and since we have many students on attachment with us sometimes the guideline books disappear. They go with them when preparing for exams" (IDI, Policy maker)

Other health system constraints that drive mistreatment include lack of sufficient space which precipitates overcrowding. This means that many young infants shared cots or incubators, exposing them to infection and interruptions of treatment or unnecessary risk:

"...Babies are placed too close to each other in the nursery, and sometimes you find another baby has removed the other baby's tube or another baby is sucking the other baby's fingers, you find that it's easy for them to infect each other with diseases." (Joint IDI, Parents)

In other instances, appropriate equipment for SYIs was unavailable or insufficient:

"You go to a facility they have an adult's Ambu bag for resuscitation. A newborn comes in [and] they continue with it even if the face mask is bigger than the baby . . . [it] could be very insensitive treatment." (IDI, Provider)

Young infants were also subjected to harsh environmental conditions. Observational data indicated that some parents were instructed to undress the infants for weighing before their turn, exposing them to cold, or young infants were cleaned with cold tap water, as hot water was only available at limited times. Parents reported unclear processes around treatment for SYIs, and lengthy and unclear discharge procedures that at times included extortion by staff. On the other hand, busy providers stated that they felt the need to prioritize critically ill SYIs, attending to discharges last:

"I have to prioritize sick babies who need my attention. Perhaps there is a baby that requires resuscitation, so you must take your time resuscitating the baby until you stabilize it. ... There are babies that need intravenous lines fixed for antibiotics, by the time you finish with them you are too exhausted, you are tired even someone who is coming to tell you "discharge my baby" ...the day will end without discharging them which is not very good, ...it's not fair, it's not economical, its expensive to retain a mother here whose baby is okay."(IDI, Provider)

In addition, inadequate supervision and laxity among providers was also reported by men:

*". . .when they are doing something bad, they just check where their superiors are and when they realize that the superior cannot see . . .. they just misbehave knowing that their superior will not be coming there any time soon. When you go to ask them something they will tell you that you are looking down on them or that you are disturbing them, that is what is causing a lot of problems in most hospitals, not just this one" (FGD, Men).*

Inadequate supervision also led to instances of non-consented care by trainee students who performed certain procedures without getting proper permission from parents. This was also reported to occur most frequently during commercial strikes or night shifts, when trainees may not be well supervised. In other situations, the influence of certain religious practices in which parents would prohibit infants from being transfused blood or receive injections influenced providers to initiate care without consent to avoid delays or worsening of conditions.

**b) Stigma and discrimination..** Some providers were observed discriminating against families of low-income status because of perceived poor personal hygiene:

*"There are those children who come from families that are not well off and they wear funny clothes, and they don't have diapers . . . so a doctor can rush to treat the patient whose baby is smart and clean, he will soothe the baby very well and help her, but when they see a baby who is shabbily dressed and is not that clean. . .because life isn't the same for everyone; you will see that he doesn't attend to that child in the right way." (FGD, Men)*

In other cases, discrimination was based on medical conditions, with reports and observations of discrimination against HIV-exposed infants who were attended to in a segregated area that was screened by curtains. Another observation was of a child with Tuberculosis who received less than optimal care and was kept separate. The nurses let the infant cry for a while without attending to him.

**c) Poor rapport between parents and providers.** A common theme across various participants was ineffective provider communication. There were several accounts from parents describing how some providers use harsh language resulting in parents feeling intimidated to ask questions about their young infant's progress.

*"When you try to ask, they feel like you are interfering with her work, or you are disturbing. She might answer you rudely" (FGD Men).*

*"Some harass you for example if you ask him a question he responds rudely" (Parent IDI)*

Other forms of poor communication were observations of minimal information provided to parents on cord care, danger signs, identifying and treating pain, and continuing medications. First-time mothers received limited information on discharge processes, breastfeeding, and general infant care. Parents stated that the ineffective communication was partly driven by their fear of providers, either due to poor provider attitudes or power dynamics where providers used harsh language or tones, leading them to avoid asking questions, seek clarification, or even properly disclosing information about the infants. Poor provider attitude led to inadequate attention to the parent, which was often expressed through negative provider body language.

*"Like some nurses are not friendly, . . . when you are attended to at a time like this when the life of your kid appears to be in the balance you need encouragement, attention, and assurance but that has not been forthcoming. You feel like you can even tell by the body language that*

*this person feels like l am bothering him or her. That is the only problem I have had" (IDI Parents).*

Fathers reported feeling isolated when not sufficiently informed or allowed to see their infants:

"*Personally, since my baby came here, I have not been allowed to see him and I have not been told what the reason is [for admission] up to now. I just know that the baby is on oxygen, he has less blood, but I don't know more up to now. They don't see the use of the father of the child being there and following up on what is going on, and I come from very far" (FGD, Men).*

Providers confirmed that there were a few cases of providers who related poorly with patients:

"*There are cases of some doctors who don't care. They just bypass patients without looking at them. They only act after the intervention of other doctors. However, such doctors are not many, it's one out of ten who are like that" (IDI, Provider)*

In general, staff shortages and workload were cited by both parents and providers as reasons for most of the negative provider-parent interaction, perceived disinterest, or non-response to parental concerns.

**d) Physically inappropriate care.**   Use of force was reported where providers slapped newborns hard to encourage them to cry soon after birth or examined newborns when they were not calm. Additional reported experiences were unnecessary repeated "pricking" while taking blood or inserting intravenous lines and other forms of handling such as rough insertion of nasogastric tubes.

"*They just do it carelessly. They handled them in an inhumane way. Or maybe when they are taking the blood samples, you see the baby is crying so much but they do not care they just inject. They were not kind to the babies." (Joint IDI, Parents).*

There were also instances where infants were not safely secured in the cots, and one observed instance where an infant fell to the ground. Other forms of physically inappropriate care were observed with multiple examples of parents trying to "forcefully" feed their young infants for fear of being threatened and verbally abused by providers if the infant did not gain adequate weight. This was observed where parents nursing infants in the Kangaroo Mother Care (KMC) rooms forcefully fed their infants hoping the infant would gain weight and be discharged faster. Insufficient feeding was observed where mothers with twins found it hard to feed more than one child in the allocated time in the newborn units:

"*Those of us who have two children, the disadvantage we have limited time to breastfeed. You find yourself struggling with one to breastfeed so that you can take the other, but the stipulated time given is not enough. You find one baby sleeping and as you struggle to wake them up you are told time is over, so you find that one child does not get milk well. So, you are asked to leave, and you are allocated the same time as someone who has only one child and we have two." (FGD, Women)*

Additionally, feeding cups used to feed sick infants were shared without proper cleaning in the rush to feed multiple infants on time and within the allocated 'feeding times':

"*In this place, the babies sleep three or four in one bed, and there are cups that we use because they can't breastfeed directly; so we express milk in the cups and then give them but sometimes the cups are not enough and you have to wait for a mother to finish and then you take the cup wash it and use it, and that can lead you to having a limited time to feed the baby.*" *(FGD, Women)*

Finally, some young infants missed feeding. This was observed when mothers were not available, and nursing staff skipped feeding infants due to their workload, especially at night.

**e) Bereavement and posthumous care.** There were varied ways in which facilities and providers responded to bereaved parents. Although some facilities have social workers or counselors, there were minimal efforts made to provide some form of emotional support or counselling when a young infant died. In some cases, there were families and parents who were able to access psychological support from the social worker or counsellor. If providers felt that the client had psychological issues, a counselor was called or the nurses themselves provided counseling. This only happened when the provider noticed extreme distress exhibited by the parents. However, overall, there was no organized form of supportive care for parents when they lost their newborns or infants. On one occasion there was no support at all.

"*So she went to check on the baby before the three hours elapsed and the nurse who was on duty that day was very harsh, first she started quarrelling her why she has gone in before time. . . Now the nurse knew the baby had died but didn't know how to approach the mother and tell her. So, when she came to ask, 'where is my baby?' the nurse did not tell her anything, she just left her standing there. That is when another nurse came and told her "Your baby didn't make it blah, blah, blah', just like that. (Joint IDI Parent)*

## Responses to consequences of mistreatment

Responses to specific mistreatment are illustrated in Fig 1 with their corresponding examples.

**Acquiescent responses.** Acquiescent responses ranged from parents feeling humiliated, resigning to the situation by not reacting for the sake of their infant's treatment, reluctantly accepting the situation, or deciding to seek care elsewhere in the future as illustrated below:

"*How does this one look, does she eat really, will this tiny baby really grow? Some of the mothers looked at me, laughed at me, and some just kept quiet, it made me angry till I cried.*" *(FGD, Women)*

"*If I take this baby to facility X, it takes so long for the baby to be seen, no adequate supplies and we are forced to buy drugs. Next time I will not seek care at [facility X], I will choose somewhere else.*" *(FGD, Men)*

Other examples of resigning to the experience were described when parents sought information about treatment procedures but received rude responses and became fearful. Providers reported feeling that parents were interfering with their work or disturbing them, leading to abrupt or unkind responses. Parents reported that these experiences caused emotional distress, likely to influence their care seeking patterns in the future, and lead to refusal to adhere to treatment:

"*A parent reported her baby removing her tube to a provider who inserted it very roughly. The baby did not feed well. The parent felt very bad and removed the tube and started cup feeding. She didn't bother with the tube again.*" *(FGD Women)*

"*I just let the providers do their work.*" *(Joint IDI Parents).*

| | Examples of mistreatment | Immediate reactions | Potential long-term consequences |
|---|---|---|---|
| **Acquiescent measures** | **Physical abuse** | Parents reported **being sad and stressed** when they witnessed their infants being pricked several times in search of a vein. | Psychological/emotional distress to the mother and infant |
| | **Harsh language or tone verbal abuse** | Parents remain **quiet to avoid conflict** so that their infants are treated and later consult another provider | Strained relationship between parent/provider |
| | | Parent was spoken to in an unfriendly way by a provider, Mother did not want to quarrel, and **instead decided to leave the facility** with her infant | Alternative care seeking patterns |
| | | A couple was rudely dismissed from a consultation room when they walked in on a provider watching a TV show. The male spouse felt **resentment towards the provider** | Strained relationship with provider |
| | **Negligence** | The provider was on the laptop and phone as the expectant woman labored, she ignored her and only reacted when the woman started bleeding. The newborn was tired and not breathing. She gave the newborn two slaps. **The woman who over bled felt hurt.** | Psychological/emotional distress |
| | **Ineffective communication to parents** | Parents are not adequately informed on procedures and treatment the infants go through. This **makes them sad and unsettled.** | Psychological/emotional distress to the parent |
| | | A lady was **angry** with how she and her infant were being treated. She had been told by doctors that her infant was fine, but she was yet to be discharged. | Strained relationship with provider |
| **Assertive measures** | **Physical abuse-** excessive pricking of infants/rough insertion of the NG Tube | A parent was annoyed and **scolded a provider** because they pricked the infant multiple times to collect blood samples without informing the parent about the tests or results. | She refused further tests leading to non-adherence to treatment |
| | | A parent reported her newborn removing her tube to a provider who inserted it very roughly. The newborn didn't feed well. The **parent felt very bad** and removed the tube and started cup feeding. **She didn't bother with the tube again**. | Strained relationship with provider |
| | **Negligence-** failure to perform full tests on infant | **Parent quarreled** with the provider in the laboratory as he failed to complete the whole test specified and the parent had to be sent back by the consulting provider to complete the tests. | Declined doing the remaining test, non-adherence to treatment |
| | **Delayed services** | Providers were in a room having informal discussions and they were very reluctant to attend to the parent's infant. The parent **scolded the provider on their laxity** and the way they approach people. | Strained relationship with provider |

**Fig 1. Responses to and consequences of mistreatment (adapted from McMahon et al.).**

**Assertive responses.** Assertive responses were rarely expressed or observed. Parents either quarreled with the provider or expressed their disappointment by becoming angry.

"*A lady was angry with how she and her baby were being treated . . . shouting along the corridor, asking them to discharge her and her baby immediately because no one was telling her why her baby was still admitted despite severally asking the doctors in the ward who kept assuring her that her baby is fine but yet she was not being discharged.*" (Notes from Observers)

An observer witnessed a parent who was annoyed and scolded a provider because they pricked the infant several times to collect blood samples without informing the parent about the tests or giving the results of the samples taken. This not only led to emotional distress but also strained the provider-parent relationship. Some parents would refuse to follow instructions given by the providers or request to be discharged, sometimes against medical advice, contributing to non-adherence to treatment.

## Discussion

This paper explores the experience of care for parents and their SYIs who sought care in newborn or pediatric units in five large hospitals in Kenya. Experiences of mistreatment of SYIs were identified in multiple categories from discussions held with a range of respondents: policy makers, providers, and parents. These reports were corroborated during ethnographic

observations and FGDs demonstrating the need to use a range of methods to describe mistreatment, poor quality of care, and contextualize findings of such a sensitive concept.

Our findings show that mistreatment of SYIs can be classified into five out of the seven categories identified by Bohren et al. for women during labor and delivery [3–5, 19] as well as those identified by Sacks for newborns [15]. Out of the seven categories identified by Bohren et al., we did not identify any incidents of sexual abuse, in concurrence with Sacks. In addition, the data did not describe any issues regarding legal accountability identified by Sacks (34). A new category that we have merged from these data: is "Physically inappropriate care." Although lack of feeding support has been classified previously under "poor rapport of providers" [15], our findings indicate that the drivers of inappropriate feeding practices were related to hospital policies indicating set feeding times, resulting in mothers complaining there was little time to breastfeed their SYIs especially when there were twins. This, along with limited spaces that cannot accommodate many parents, issues of poor communication by providers, and lack of enough equipment (such as sterile feeding cups) suggest that this could be a category of mistreatment on its own. In addition to trying to ensure the SYIs get the required feeds, the restrictive feeding times prevent bonding and can impact the mother-infant relationship [16].

Another aspect of physically inappropriate care encompasses physical and verbal abuse which was perceived to be directed at both the SYI and the parent. Most reports of physical abuse described providers handling the infant roughly, and parents encountered providers who verbally responded rudely and harshly to them. This contributed to feeling that there was poor rapport between parents and providers resulting in a lack of communication such that parents were unaware of what was happening to their young infant. Some fathers described not knowing anything about their sick infant and others had not had an opportunity to see their newborn for the first time because of lack of access to the neonatal unit. Some providers 'blamed' parents for bringing their SYI late.

Several of the classification for mistreatment are also quality of care issues and directly relate to the WHO pediatric quality of care framework–especially from the experience of care domain–and the standards of care for the small and sick newborn [13]. For example, Standard 4 outlines communication with small and sick newborns and their families is effective, with meaningful participation, and responds to their needs and preferences, and parental involvement is encouraged and supported throughout the care pathway. This standard is linked to the mistreatment category of poor provider-parent rapport. Standard 7 for small and sick newborn requires availability of competent, motivated, and empathetic staff and Standard 8 indicates that each facility has appropriate physical environment for routine care and management of complications in small and sick newborns. When analyzing the data in the context of these standards, it becomes clear that the consequences of poor quality of care can manifest as mistreatment. The data describe numerous examples of failures to meet professional standards of care. This includes parents reporting that providers did not request informed consent prior to any procedures on their infant as well as neglect and abandonment whereby nurses did not respond to urgent requests by parents. Although there is limited literature describing this in infants, the maternal health literature has described this [20, 21]. https://journals.plos.org/plosone/article?id=10.1371/journal.pone.0229923 - pone.0229923.ref004Our data also reinforce the well-established understanding that various forms of mistreatment are a result of health system conditions and constraints [22]. A complex range of systemic failures at health system, facility, and individual levels contribute to poor quality of care [3, 19]. The health system failures include inadequate essential resources, insufficient number of skilled providers, limited equipment, supplies, and lack of bed space to manage sick infants. These drivers were common and have been reported elsewhere [23–25]. However, the impact of the environment

on the development of the brain and possible long-term outcomes of poor quality of care environments is not often considered.

Poor governance of the health system has also been identified as a driver of mistreatment. At the facility level, poor managerial oversight and weak accountability measures may contribute to a poor environment for health workers, a feature which is common in many settings [21, 26]. At the individual level, limited provider knowledge, negative attitudes, and low motivation contribute to poor provision of care and neglect [25].

With regard to stigma and discrimination, Bohren et al. identified four categories: 1) ethnicity/race/religion, 2) age, 3) socio-economic status and 4) medical conditions [2]. Most parents reported that poorer, less well-dressed women and those with HIV were more often ignored or had to wait longer to be seen. However, it was not clear whether the age of parents or participant ethnicity increased the likelihood of mistreatment in our study.

The inadequate support given to parents during bereavement indicates the need for stronger structures for psycho-social support for parents, especially when their SYI has a prolonged hospital stay or their infant dies. Strengthening support structures and internal mechanisms of accountability, such as appropriate reporting and feedback systems for resolving mistreatment, will improve overall quality of care and address manifestations of mistreatment in bereavement and posthumous care [15].

By examining the effect of mistreatment on the parents of hospitalized SYIs, it was clear that parents' reactions to experiencing their infants' mistreatment mirror what has been documented for women during childbirth. Parents described being resigned to the circumstances or retaliated by scolding the providers. It is possible that women sometimes express agency through other types of negotiations and control, which may appear as acquiescing, but may not necessarily be the case. For example, our study shows that women felt humiliated and resigning to the situation which is an adaptive behavior that could suggest resistance strategy to mitigate providers' disrespectful treatment to their children. This has been documented in Ghana where such a strategy help women evade public humiliation because of inadequate privacy in the hospitals which affect decision-making and care provision [27]. These measures have also been documented elsewhere as acceptance and forgiveness or retaliation against the provider [23, 28]. Additionally, our data illustrate both immediate responses, such as failure to adhere to recommended treatment procedures, and long-term effects, such as changes to future care-seeking behaviors. Documenting parents' experiences and their responses to their infants' mistreatment has illuminated a potential pathway of effects of mistreatment. We argue that interventions should not only address the drivers of mistreatment, but also ensure that effects of mistreatment do not have lasting implications for the young infant or parent even after discharge from the hospital.

Fig 2 also shows that the experience of mistreatment is likely to not only affect future care seeking, including for extended postnatal care, which is largely underutilized, but may also have emotional effects that could have a negative impact on the parents as they continue to care for a vulnerable infant. Although we present a linear pathway for the sake of simplicity, the relationship between drivers of mistreatment, manifestations, and the corresponding effects are complex. However, the linear pathway helps to illuminate potential areas of interventions to address not only the underlying drivers but also the consequences of mistreatment. Finally, our data demonstrate that parents often suffer the experience of mistreatment projected on their SYIs, which affects their emotional wellbeing.

Attempts to improve experience of care for SYI requires the recognition and application of a person-centered care approach together with a family-centered approach [29] to ensure SYIs receive quality age-appropriate care. This includes developing interventions that address provider-parent communication, including emotional needs of parents as well as supporting them to be engaged in the care of their infants while in hospital [16]. Moreover, encouraging

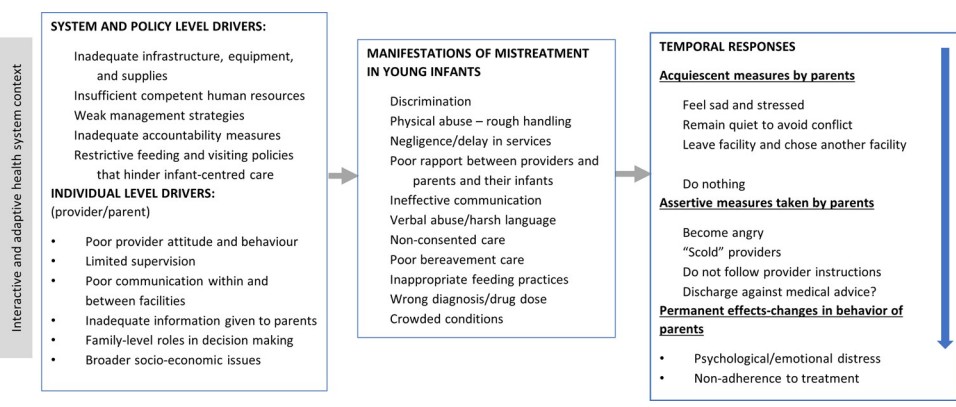

**Fig 2. Pathways of mistreatment and potential consequence.**

newborn units to identify solutions to their localized health system challenges may contribute to a more supportive work environment for all providers [30].

## Strengths and limitations

Despite challenges of data collection such as getting appointments with men and potential biases associated with observations, the use of multiple methods (observations and interviews) and documentation of in-depth information may have mitigated this effect. We only collected data in large five hospitals which may not necessarily reflect experiences in all hospitals in Kenya. However, these experiences may be applicable to secondary and tertiary institutions, which have similar characteristics. This study focused on experiences of care for SYIs and their parents in hospitals. Therefore, other types of mistreatment, including that experienced by healthy newborns, and experiences at the community level, are not included. This analysis is also limited to experiences of SYIs up to 59 days.

## Conclusion

This study outlines types of mistreatment that were observed or reported for SYIs in five hospitals in Kenya and explores the responses and consequences of parents. Mistreatment for SYIs appears to be prevalent and linked to poor quality of care. To address mistreatment in this group of very young children, interventions that focus on building better communication, responding to the developmental needs of infants and emotional needs for the parents, strengthen the number of providers and their competencies in newborn care, as well as a supportive, enabling, and healthy environments, will lead to more respectful quality care for newborns and young infants.

## Acknowledgments

We thank all the study participants, policy makers, providers, and parents of sick young infants who narrated their experiences in either providing services or receiving services for their infants. Thanks to the Ministry of Health, County Health Department, and Facility Managers who provided us with the opportunity to conduct the study.

## Author Contributions

**Conceptualization:** Timothy Abuya, Charlotte E. Warren.

**Formal analysis:** Timothy Abuya, Charity Ndwiga, Chantalle Okondo, Emma Sacks, Pooja Sripad.

**Funding acquisition:** Charlotte E. Warren.

**Investigation:** Timothy Abuya, Charity Ndwiga, Chantalle Okondo, Pooja Sripad.

**Project administration:** Charlotte E. Warren.

**Writing – original draft:** Timothy Abuya.

**Writing – review & editing:** Timothy Abuya, Charlotte E. Warren, Charity Ndwiga, Chantalle Okondo, Emma Sacks, Pooja Sripad.

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
