## [Decision Letter · Decision Letter 0]

11 Jun 2021

PONE-D-20-28509

Manifestations and drivers of mistreatment of sick newborns and young infants and their parents in health facilities in Kenya

PLOS ONE

Dear Dr. Abuya,

Thank you for submitting your manuscript to PLOS ONE. After careful consideration, we feel that it has merit but does not fully meet PLOS ONE’s publication criteria as it currently stands. Therefore, we invite you to submit a revised version of the manuscript that addresses the points raised during the review process.

ACADEMIC EDITOR: 

The reviewers have recommended some revisions to your paper. Please address these in a revised manuscript particularly considering the use of a framework to present the qualitative findings and including some recommendations for suggested action.

We look forward to receiving your revised manuscript.

Kind regards,

Tanya Doherty, PhD

Academic Editor

PLOS ONE

Journal Requirements:

2. Please clarify in your ethics statement whether you received ethics approval from an ethics committee in Kenya to conduct this study, and whether the hospitals where the research took place approved the study.

4. We note you have included a table to which you do not refer in the text of your manuscript. Please ensure that you refer to Table 2 in your text; if accepted, production will need this reference to link the reader to the Table.

5. Please include a copy of Table 3 which you refer to in your text on page 17.

Reviewers' comments:

Reviewer's Responses to Questions

**Comments to the Author**

1. Is the manuscript technically sound, and do the data support the conclusions?

Reviewer #1: Partly

Reviewer #2: Partly

Reviewer #3: Yes

2. Has the statistical analysis been performed appropriately and rigorously? 

Reviewer #1: Yes

Reviewer #2: No

Reviewer #3: I Don't Know

3. Have the authors made all data underlying the findings in their manuscript fully available?

Reviewer #1: Yes

Reviewer #2: No

Reviewer #3: Yes

4. Is the manuscript presented in an intelligible fashion and written in standard English?

Reviewer #1: Yes

Reviewer #2: Yes

Reviewer #3: Yes

5. Review Comments to the Author

Reviewer #1: This paper addresses the issue of mistreatment of newborns and young infants in health facilities in Kenya. It is an important topic considering global and national efforts to expand newborn care services in low- and middle-income countries for reaching the sustainable development goal targets. The authors define categories of mistreatment and the response of parents to it and suggest approaches to address the drivers of mistreatment.

While the paper is interesting and well written, I think it can be further improved by including more information that justify this classification of mistreatment, especially in relation to related literature on quality of newborn care services and on its impact on neonatal health outcomes. In addition, the conclusions can be improved by mentioning human resources for health strategies. With modifications addressing the detailed comments below, and expanding on the use of other frameworks (quality of care, nurturing care, newborn rights), this could be an influential paper.

Detailed comments are provided below by section.

1. Abstract

1.1. While it is not common to find papers on mistreatment of newborns and young infants in health facilities, there is a body of literature on quality of newborn care services and on newborn experience of care, especially in relation with development. I would rephrase the statement on "limited efforts to understand the experience of care for sick newborns and young infants".

1.2 In the conclusions the authors should include a recommendation on strategies to strengthen human resources.

2. Introduction

2.1 Page 9, "Negative experiences during labor and childbirth are a deterrent to the use of skilled birth services"...It would better "deterrent to the choice of birth in health facilities". In principle "skilled birth services" should be respectful by definition.

2.2 Page 10, the authors should mention here, after reference 12, or in place of reference 12, the Standards for improving the quality of care for small and sick newborns in health facilities, WHO 2020, which articulate the concept of experience of care from the newborn perspective with three standards focusing on the needs and rights of newborns and their families all along the care process.

2.3 Page 10, see comment 1.1. In addition, the authors may consider adding a sentence on the fact that newborns and young infants express their experience of care with signs of distress or calm depending on the sensorial environment in which they are cared for, the importance of the caring environment and how this can impact on the development of the brain, and therefore the importance of providing respectful and nurturing care, care that responds to the needs of the infant, because of the long lasting impact on the infant, not only on parents. Please refer to the literature on infant and family-centered developmental care, kangaroo mother care and nurturing care.

3. Methods

Overall the methods looks sound. More information on coding framework would be helpful.

4. Results

4.1 I understand that the classification of mistreatment is based on previous work on mistreatment in maternal health and in the newborn in the postnatal period. However, over the past five years there has been an evolution in the concept of disrespect and mistreatment which led to an update of the Respectful Maternity Care Charter to include the newborn in 2019 and to the publication by WHO of the Standards for improving the quality of care for small and sick newborns in health facilities in 2020. In fact the categories of mistreatment correspond to poor quality of care as well. Therefore, it would interesting to see a link with those documents. Also some of the categories reflect more the quality of interaction with parents than the infant. Please note that all categories of mistreatment presented in the paper are reflected in the new newborn standards and perhaps the quality of care framework of the standard with its 8 domains represents a better way of categorizing the results, in the sense of offering a way of focusing only on infant quality of care and reducing overlaps between groups.

4.2 Page 14

There is a slight difference between failure to meet professional standards and failure to meet standards of care in the sense that the first is more related to staff knowledge, attitudes and behaviors while the second is related not only to human resources challenges but also to other health system constraints.

4.3 Page 14 line 8

Are the authors intending "or" or "and"? Please clarify if failure to meet professional standards is interpreted as a staff behavior mostly related to heavy workload or not.

It looks like many other reasons, including poor organization of care, contributed to delays in care provision.

4.5 Page 14, overcrowding example. This seems to be an example of failure to meet standards of care more than professional standards. It seems to fit better among health system constraints.

4.6 Page 15 , last paragraph. These examples seem to fit better in the previous category: failure to meet professional standards, as they are related to human resources competence.

4.6 Page 19. Please explain what is intended here for forcefully: more often than prescribed, higher quantity than prescribed, which feeding technique, how babies reacted, etc.

4.7. Page 20. The authors may consider having verbal abuse included under "Poor rapport between parents and providers" instead of having it as a separate category or consider having a category on poor communication, including verbal abuse, and all the other issues as a separate category named "Disrespect and poor interaction between providers and parents", to make the categories more pertinent to newborn and infant care.

4.8. Page 22. I would classify refuse to follow instructions and request for discharge as an assertive response, given the context of Kenya.

5. Discussion

5.1 As already mentioned the issues highlighted in the paper are quality of care issues. The authors should expand the discussion. For example, if the current classification of mistreatment is maintained, the discussion will benefit from showing how the categories of maltreatment relate to the WHO quality of care framework and the standards of care for small as sick newborn, particularly those on the experience of care.

Just as an example: category 1 relates closely to provision of care standard statements S1.2. S1.3 and S1.4; category 2 relates to standard statements S1.36, S1.40, S1.41, as well as standards 7 and 8; category 3 relates to standard statements S.1 to S.4; category 4 can be linked to standard statements S5.1, S5.2, S5.3, and S6.4; category 5 is linked to standard statements S1.8, S1.9 and 5.4; category 7 to standard statements S4.1 to S4.6 which details communication; and category 8 can be linked to standard statement S5.6 among others.

5.2 Consider adding some considerations on the concept of person centered care and parents participation in the care of their infant.

5.3 Consider adding a paragraph on the impact of the environment on the development of the brain and possible long term outcomes of poor quality of care environments.

6. Conclusions

The authors should consider strengthening the conclusions with a statement on the importance of putting in place strategies to strengthen human resources for newborn care, number and competencies, as well as working environment.

Reviewer #2: The article aimed about an interesting topic however the analysis and findings do not support to answer a construct comprising of themes that would answer your research questions given as objective. I believe that the findings under what you call as themes are just summaries of the topics that were found within the coded text. "1) failure to meet professional standards manifested as abandonment and delayed provision of care; 2) health system conditions and constraints; 3) limited provider skills; 4) stigma and discrimination due to provider perception of personal hygiene or medical condition; 5)

physical abuse: providers take blood samples and insert intravenous lines and gastric tubes in a rough

manner; 6) inappropriate feeding practices: parents forcefully feed infants to avoid providers’ anger or share

unsterile feeding cups; 7) poor parental-provider rapport expressed as ineffective communication, perceived

disinterest, and non-consented care; and 8) no organized form of bereavement and posthumous care in the case

of infant’s death". The analysis in qualitative research is more than that. I will suggest that you give your analysis a review from a qualitative researcher to identify three or a maximum of four themes. Some of the current themes for example are not themes, such as: failure to meet professional standards manifested as abandonment and delayed provision of care. This is related to access to care and touches upon the three delay model of maternal care. Without keeping the findings in the existing frameworks of access to care, mistreatment of patients in low and middle income countries the study will not have a strength and the results may not be useful for wider audience. The figures included are merely summaries of the data that has already been presented.

The research aim: "This paper describes what constitutes mistreatment of sick newborns and sick young infants (SYIs), drivers, responses, and the effects of these experiences on parents." is not focused and lacks has several items put together.

Are focused group discussions with women and men useful to understand what mistreatment they received during their care or the care of their children. I think not. Please re consider the methods.

Reviewer #3: This manuscript is well written with a clear description of the design, methods, ethical considerations, results, and conclusions. The tables are especially helpful in summarizing the results. This manuscript was emotionally difficult to read but made a compelling case regarding the need for improvements in providing medical care that is compassionate and humane for sick babies and their parents. I understand the purpose of this article was to describe a problem, but I would also like to see some suggested solutions or at least a hint that solutions will be sought. Perhaps a follow-up article will describe an action plan for seeking improvements, which will need to be comprehensive as the problems seem to be so multifaceted. Effective solutions must address the need for multidisciplinary education about technical medical skills, special needs of infants and families, compassionate attitudes, and respectful communication, as well as the obvious need for adequate staffing and equipment. A summary of the comprehensive needs required to address this issue would be welcomed in the Discussion or Conclusion sections of this manuscript. If that is thought to be beyond the scope of this manuscript, I will look forward to a follow-up manuscript to address this important topic. Thank you for bringing focus to these issues that are of international significance, as I strongly suspect these problems are not limited to 5 hospital in Kenya.

6. PLOS authors have the option to publish the peer review history of their article (what does this mean?). If published, this will include your full peer review and any attached files.

Reviewer #1: **Yes: **Ornella Lincetto

Reviewer #2: No

Reviewer #3: **Yes: **Raylene M. Phillips, MD, MA, FAAP, FABM, IBCLC

---

## [Author Response · Author response to Decision Letter 0]

5 Aug 2021

Changes made to the paper.

Editors’ comments and changes made 

Comment 1: Please clarify in your ethics statement whether you received ethics approval from an ethics committee in Kenya to conduct this study, and whether the hospitals where the research took place approved the study.

Response 1: This has been updated in the section of ethics.

 Comment 2: We note you have included a table to which you do not refer in the text of your manuscript. Please ensure that you refer to Table 2 in your text; if accepted, production will need this reference to link the reader to the Table. Please include a copy of Table 3 which you refer to in your text on page 17.

Response 2: We recognize this was an error and has since been corrected

Comments from Reviewer 1:

Comment 1: This paper addresses the issue of mistreatment of newborns and young infants in health facilities in Kenya. It is an important topic considering global and national efforts to expand newborn care services in low- and middle-income countries for reaching the sustainable development goal targets. The authors define categories of mistreatment and the response of parents to it and suggest approaches to address the drivers of mistreatment.

While the paper is interesting and well written, I think it can be further improved by including more information that justify this classification of mistreatment, especially in relation to related literature on quality of newborn care services and on its impact on neonatal health outcomes. In addition, the conclusions can be improved by mentioning human resources for health strategies. With modifications addressing the detailed comments below and expanding on the use of other frameworks (quality of care, nurturing care, newborn rights), this could be an influential paper.

Response 1: We thank the reviewer for this comment. we have made the appropriate amends in the paper taking account of these reflections. We present a detailed account of changes in each specific comments below. 

Detailed comments are provided below by section.

1. Abstract

Comment 1.1. While it is not common to find papers on mistreatment of newborns and young infants in health facilities, there is a body of literature on quality of newborn care services and on newborn experience of care, especially in relation with development. I would rephrase the statement on "limited efforts to understand the experience of care for sick newborns and young infants".

Response 1.1 The abstract has been amended with this comment in mind and now it reads: 

“Despite efforts to incorporate experience of care for women and newborns in global quality standards, there have been limited efforts to understand experience of care for sick newborns and young infants (aged 0-59 days). This paper describes what constitutes mistreatment of sick young infants (SYIs), drivers, and parental responses in hospital settings in Kenya”. 

Comment 1.2: In the conclusions the authors should include a recommendation on strategies to strengthen human resources.

Response 1.2: We have included potential strategies for both clients and providers. For providers we have stated that there is need to have supportive environment including emotional needs. The conclusion section of the abstract now reads: 

“Conclusion: Mistreatment of newborns and SYI is common and requires strategies that address drivers and effects of mistreatment. Approaches that build better communication, address the emotional needs of parents and providers, and provide supportive, enabling, and healthy environments will lead to more respectful care of newborns and young infants”

2. Introduction

Comment 2.1 Page 9, "Negative experiences during labor and childbirth are a deterrent to the use of skilled birth services"...It would better "deterrent to the choice of birth in health facilities". In principle "skilled birth services" should be respectful by definition.

Response 2.1: This has been amended and is reflected in page 3 of the current article. 

Comment 2.2 Page 10, the authors should mention here, after reference 12, or in place of reference 12, the Standards for improving the quality of care for small and sick newborns in health facilities, WHO 2020, which articulate the concept of experience of care from the newborn perspective with three standards focusing on the needs and rights of newborns and their families all along the care process.

Response 2.2: We have made additions to the stated paragraph by linking the overall standards for maternal health to the WHO published standards for improving quality of care for small and sick newborns. The part now reads:

“Attention to the experience of care of the newborn is more recent. The Respectful Maternity Care (RMC) Charter, published in 2011 and updated in 2019, articulates 10 fundamental rights of childbearing women and newborns. Inclusion of the newborn was essential to provide a framework for understanding the combined mother and newborn experience of care (11). The World Health Organization (WHO) released a statement on the elimination of disrespect and abuse of women in childbirth in 2014 (7) and subsequently a framework for quality of care for maternal and newborn health (MNH) in 2016 (12). The WHO MNH quality of care framework comprises eight domains including both the provision of care for and the experience of care by women and newborns in health facilities. Experience of care consists of effective provider communication with women and their families about the care provided, parents’ expectations and rights, care with respect and preservation of dignity, and access to social and emotional support for care received or events that may present during care. The cross-cutting areas of both experience and provision of care include availability of competent, motivated human resources and the physical resources that are prerequisites for good quality (12). More recently, WHO published standards for improving the quality of care for small and sick newborns in health facilities which articulate the concept of experience of care from the newborn perspective with three standards focusing on the needs and rights of newborns and their families all along the care process (13).”

Comment 2.3 Page 10, see comment 1.1. In addition, the authors may consider adding a sentence on the fact that newborns and young infants express their experience of care with signs of distress or calm depending on the sensorial environment in which they are cared for, the importance of the caring environment and how this can impact on the development of the brain, and therefore the importance of providing respectful and nurturing care, care that responds to the needs of the infant, because of the long lasting impact on the infant, not only on parents. Please refer to the literature on infant and family-centered developmental care, kangaroo mother care and nurturing care.

Response 2.3: We appreciate the reviewer’s comments and have made amends to the stated paragraph to reflect these thoughts. Two we have referenced one the seminal papers on nurturing care: The section now reads

“The complexity of conceptualizing experience of care by newborns and young infants is compounded by their inability to verbally express their needs or share their experiences (15). Newborns and young infants express their experience of care with signs of distress or calm depending on the sensorial environment, indicating the importance of the caring environment and how this can impact on their development. Due to the long-lasting impact on the infant, there is need for providing respectful and nurturing care that responds to the needs of the infant(16). Therefore, this paper explores the experience of care of parents—mothers and fathers (or other close relatives)—who sought care for their sick young infants (SYIs) in five large hospitals in Kenya. We describe what constitutes mistreatment of SYIs (all sick babies 0-59 days old), what drives these provider behaviors, parents’ immediate responses to mistreatment of SYIs, and the effects of these experiences on parents” 

3. Methods

Comment 3.1: Overall the methods look sound. More information on coding framework would be helpful.

Response 3.1: We have made clarifications to the process of coding to include the number of themes and how we adapted the existing framework to our analytical process. Details of the changes are reflected in the last paragraph in page seven, and it reads:

“Thematic analysis, adapting existing mistreatment frameworks(3, 15), guided our analytical approach. Types of mistreatment that parents of SYIs experienced and understood or observed happening to SYIs were categorized using the typology described by Bohren et al. for maternal health (3) and expanded by Sacks to include newborns (15). A team of 3 researchers reviewed transcripts, field notes and observations, making initial annotations around topics arising from the data. An initial codebook of 19 themes and 100 sub-themes was developed and refined based on the process of open coding and progressive categorization of issues that emerged. Transcripts were exported into Nvivo 12 (QSR International) software with facility names redacted and coded by 3 researchers. This was followed by a process of summarizing mistreatment types experienced by SYIs and their parent’s responses to this negative experiences. This process enabled the research team to explore similarities and differences in mistreatment and responses across sites and by SYI age category (between 0-59 days). The deliberative process looking at summaries in relation to the extant literature led to themes aggregated in categories that aligned closely with existing frameworks used to describe mistreatment of women during childbirth and the subsequent responses (3, 15, 18). We adapted these categories to the SYI population during the interpretation and presentation of our results. For example, we used responses to mistreatment described by McMahon (18) and expanded it to extrapolate their effects categorized as short-term and long-term consequences. The responses were either acquiescent measures, which are non-confrontational methods, or assertive measures, which are more confrontational methods to try to address mistreatment.

4. Results

Comment 4.1 I understand that the classification of mistreatment is based on previous work on mistreatment in maternal health and in the newborn in the postnatal period. However, over the past five years there has been an evolution in the concept of disrespect and mistreatment which led to an update of the Respectful Maternity Care Charter to include the newborn in 2019 and to the publication by WHO of the Standards for improving the quality of care for small and sick newborns in health facilities in 2020. In fact, the categories of mistreatment correspond to poor quality of care as well. Therefore, it would be interesting to see a link with those documents. Also some of the categories reflect more the quality of interaction with parents than the infant. Please note that all categories of mistreatment presented in the paper are reflected in the new newborn standards and perhaps the quality-of-care framework of the standard with its 8 domains represents a better way of categorizing the results, in the sense of offering a way of focusing only on infant quality of care and reducing overlaps between groups.

Response 4.1: We appreciate this observation, and we recognize the value of linking our results to the existing standards. Since we adapted the framework in developing typologies, we have made this clear in the methods section. Additionally, we have clarified in the discussion section how these results fit with the WHO standards of newborn care. In view of this, we have retained the structure of the results with clarity of how we have used the framework. Wherever there are overlaps we have attempted to illustrate that using examples and comment that there are categories that overlap but the essence will be how best to address the underlying drivers to minimize the observe and experience mistreatment. 

Comment 4.2 Page 14

There is a slight difference between failure to meet professional standards and failure to meet standards of care in the sense that the first is more related to staff knowledge, attitudes and behaviors while the second is related not only to human resources challenges but also to other health system constraints.

Response 4.2: This is well thought out reflections. we have made the necessary adjustments to the section to incorporate this comments. The section now reads: 

a) Failure to meet professional standards and health system conditions and constraints 

In adapting the Sacks framework, we distinguish between failure to meet professional standards by providers and the health system ability to provide adequate standards of care. The former is related to staff knowledge, attitudes, and behaviors while the latter is related to the health system constraints which then drive mistreatment. The complexity of this interaction illustrates the challenges of describing discrete categories of manifestations and what drives them. For example, both parents and providers reported examples of failure to meet professional standards during care of SYIs. Observations and parents’ reports indicated that at times, providers took many breaks contributing to long queues of patients waiting to be seen. Another example of failure to meet professional standards by providers was reports of abandonment and neglect manifested through providers’ failure to monitor treatment procedures or provide timely care because of poor organization of care, contributing to delays in care provision. These were also cited as driven by heavy workload of providers - a wider health system constraint:.”

Comment 4.3 Page 14 line 8

Are the authors intending "or" or "and"? Please clarify if failure to meet professional standards is interpreted as a staff behavior mostly related to heavy workload or not.

It looks like many other reasons, including poor organization of care, contributed to delays in care provision.

Response 4.3: This has been adjusted see responses to comment 4.2 above

Comment 4.5 Page 14, overcrowding example. This seems to be an example of failure to meet standards of care more than professional standards. It seems to fit better among health system constraints.

Comment 4.6 Page 15 , last paragraph. These examples seem to fit better in the previous category: failure to meet professional standards, as they are related to human resources competence.

Response 4.5 and 4.6: This section has been amended see response 4.2 above. 

Comment 4.6 Page 19. Please explain what is intended here for forcefully: more often than prescribed, higher quantity than prescribed, which feeding technique, how babies reacted, etc.

Response 4.6: We observed cases of mothers trying to feed their children multiple times with a desire and a perception that this will help their children to add weight faster as part of a response to avoid scolding from providers who often abused mothers of those whose children do not add weight as expected. 

Comment 4.7. Page 20. The authors may consider having verbal abuse included under "Poor rapport between parents and providers" instead of having it as a separate category or consider having a category on poor communication, including verbal abuse, and all the other issues as a separate category named "Disrespect and poor interaction between providers and parents", to make the categories more pertinent to newborn and infant care.

Response 4.7: This has been adjusted and is now described under the category of poor rapport between parents and providers. 

Comment 4.8. Page 22. I would classify refuse to follow instructions and request for discharge as an assertive response, given the context of Kenya.

Response 4.8: this has been amended and has been described under assertive section. 

5. Discussion

Comment 5.1 As already mentioned the issues highlighted in the paper are quality of care issues. The authors should expand the discussion. For example, if the current classification of mistreatment is maintained, the discussion will benefit from showing how the categories of maltreatment relate to the WHO quality of care framework and the standards of care for small as sick newborn, particularly those on the experience of care. Just as an example: category 1 relates closely to provision of care standard statements S1.2. S1.3 and S1.4; category 2 relates to standard statements S1.36, S1.40, S1.41, as well as standards 7 and 8; category 3 relates to standard statements S.1 to S.4; category 4 can be linked to standard statements S5.1, S5.2, S5.3, and S6.4; category 5 is linked to standard statements S1.8, S1.9 and 5.4; category 7 to standard statements S4.1 to S4.6 which details communication; and category 8 can be linked to standard statement S5.6 among others.

Response 5.1: We thank the reviewer for this observation. We recognize the how the categories we describe are closely linked to the quality-of-care issues and provision of care standards for small and sick newborn. We have used a couple of examples of the standards in the discussion and have reflected on its applicability to our results. This has been expanded in page 18 of the article in the discussion section which reads: 

“Several of the classification for mistreatment are also quality of care issues and directly relate to the WHO pediatric quality of care framework – especially from the experience of care domain – and the standards of care for the small and sick newborn(13). For example, Standard 4 outlines communication with small and sick newborns and their families is effective, with meaningful participation, and responds to their needs and preferences, and parental involvement is encouraged and supported throughout the care pathway. This standard is linked to the mistreatment category of poor provider-parent rapport. Standard 7 for small and sick newborn requires availability of competent, motivated, and empathetic staff and Standard 8 indicates that each facility has appropriate physical environment for routine care and management of complications in small and sick newborns. These standards when applied to this data illustrate consequences of poor quality of care which manifest as mistreatment. Another example is when on numerous occasions, the data describe failures to meet professional standards of care. This includes parents reporting that providers did not request informed consent prior to any procedures on their infant as well as neglect and abandonment whereby nurses did not respond to urgent requests by parents. Although there is limited literature describing this in infants, the maternal health literature has described this (20, 21). Our data also reinforce the well-established understanding that various forms of mistreatment are a result of health system conditions and constraints (22). A complex range of systemic failures at health system, facility, and individual levels contribute to poor quality of care (3, 19). The health system failures include inadequate essential resources, insufficient number of skilled providers, limited equipment, supplies, and lack of bed space to manage sick infants. These drivers were common and have been reported elsewhere (23-25). However, the impact of the environment on the development of the brain and possible long-term outcomes of poor quality of care environments is not often considered”

Comment 5.2 Consider adding some considerations on the concept of person-centered care and parents participation in the care of their infant.

Response 5.2: We note this response and have inserted several suggestions on how to incorporate person entered care. For example, in in page 20 the second paragraph reads:

“Documenting parents’ experiences and their responses to their infants’ mistreatment has illuminated a potential pathway of effects of mistreatment. We argue that interventions should not only address the drivers of mistreatment, but also ensure that effects of mistreatment do not have lasting implications for the young infant or parent even after discharge from the hospital” 

 Another example is in page 21 which includes specific interventions that can be implemented: 

“Attempts to improve experience of care for SYI requires the recognition and application of a person-centered care approach together with a family-centered approach(28) to ensure SYIs receive quality age-appropriate care. This includes developing interventions that address provider – parent communication, including emotional needs of parents as well as supporting them to be engaged in the care of their infants while in hospital (16). Moreover, encouraging newborn units to identify solutions to their localized health system challenges may contribute to a more supportive work environment for all providers(29)” 

Comment 5.3 Consider adding a paragraph on the impact of the environment on the development of the brain and possible long-term outcomes of poor quality of care environments.

Response 5.3: This has been reinforced in page 19 of the article and reads: “Our data also reinforce the well-established understanding that various forms of mistreatment are a result of health system conditions and constraints (22). A complex range of systemic failures at health system, facility, and individual levels contribute to poor quality of care (3, 19). The health system failures include inadequate essential resources, insufficient number of skilled providers, limited equipment, supplies, and lack of bed space to manage sick infants. These drivers were common and have been reported elsewhere (23-25). However, the impact of the environment on the development of the brain and possible long-term outcomes of poor quality of care environments is not often considered”

6. Conclusions

The authors should consider strengthening the conclusions with a statement on the importance of putting in place strategies to strengthen human resources for newborn care, number and competencies, as well as working environment.

Response 6: We thank the reviewer for this. we have made amends to strengthen the conclusion which incorporates the need to strengthen human resources and emotional needs for parents and providers as well: it now reads: “This study outlines the types of mistreatment that were observed or reported for SYIs in five hospitals in Kenya and explores the responses and consequences on parents. Mistreatment for SYIs appears to be prevalent and linked to poor quality of care. To address mistreatment in this group of very young children, interventions that build better communication, address emotional needs for the parents, strengthen the number of providers and their competencies in newborn care, as well as a supportive, enabling, and healthy environments will lead to more respectful quality care for newborns and young infants”

Comments from Reviewer 2:

Reviewer #2: The article aimed about an interesting topic however the analysis and findings do not support to answer a construct comprising of themes that would answer your research questions given as objective. I believe that the findings under what you call as themes are just summaries of the topics that were found within the coded text. "1) failure to meet professional standards manifested as abandonment and delayed provision of care; 2) health system conditions and constraints; 3) limited provider skills; 4) stigma and discrimination due to provider perception of personal hygiene or medical condition; 5) physical abuse: providers take blood samples and insert intravenous lines and gastric tubes in a rough manner; 6) inappropriate feeding practices: parents forcefully feed infants to avoid providers’ anger or share unsterile feeding cups; 7) poor parental-provider rapport expressed as ineffective communication, perceived disinterest, and non-consented care; and 8) no organized form of bereavement and posthumous care in the case of infant’s death". 

Comment 2.1 The analysis in qualitative research is more than that. I will suggest that you give your analysis a review from a qualitative researcher to identify three or a maximum of four themes. Some of the current themes for example are not themes, such as: failure to meet professional standards manifested as abandonment and delayed provision of care. This is related to access to care and touches upon the three-delay model of maternal care. Without keeping the findings in the existing frameworks of access to care, mistreatment of patients in low- and middle-income countries the study will not have a strength and the results may not be useful for wider audience. The figures included are merely summaries of the data that has already been presented.

Responses to reviewer 2

We appreciate this comment. We have amendments to the method sections to illustrate three things. First, we have provided a detailed explanation of the frameworks that we drew on that guided our interpretation of the data in the methods section. Two, we believe that the layered analysis draws from basic coding of themes to exploring connections with existing frameworks of mistreatment such as those described by Bohren et all and responses to mistreatment by McMahon et all. Three, we have demonstrated that although we don’t present all the themes in this paper, we have captured the typologies that speak to the framework we adapted for the analytical process. However, we do state the number of themes that were generated from the analysis process and presented only the relevant typology for the focus area of this paper. 

We hope that with the revision the process of post hoc interpretation following inductive coding approach is amenable. 

Comment 2.2: The research aim: "This paper describes what constitutes mistreatment of sick newborns and sick young infants (SYIs), drivers, responses, and the effects of these experiences on parents." is not focused and lacks has several items put together.

Response 2.2: We have revised our aim to be clearer about what the focus is. It now reads in page 4 of the main paper as follows: “Therefore, this paper explores the experience of care of parents—mothers and fathers (or other close relatives)—who sought care for their sick young infants (SYIs) in five large hospitals in Kenya. We describe what constitutes mistreatment of SYIs (all sick babies 0-59 days old), what drives these provider behaviors, parents’ immediate responses to mistreatment of SYIs, and the consequences of these experiences on parents”

This has also been reflected in the tile which reads “Manifestations, responses and consequences of mistreatment of sick newborns and young infants and their parents in health facilities in Kenya: 

Comment 2.3 Are focused group discussions with women and men useful to understand what mistreatment they received during their care or the care of their children. I think not. Please re consider the methods.

Response 2.3: Thanks for this comment – we have clarified in the methods why FGDs were used – as ways of describing norms of mistreatment and parental responses in hospital setting.

Comments from reviewer 3

Reviewer #3: This manuscript is well written with a clear description of the design, methods, ethical considerations, results, and conclusions. The tables are especially helpful in summarizing the results. This manuscript was emotionally difficult to read but made a compelling case regarding the need for improvements in providing medical care that is compassionate and humane for sick babies and their parents. I understand the purpose of this article was to describe a problem, but I would also like to see some suggested solutions or at least a hint that solutions will be sought. Perhaps a follow-up article will describe an action plan for seeking improvements, which will need to be comprehensive as the problems seem to be so multifaceted. Effective solutions must address the need for multidisciplinary education about technical medical skills, special needs of infants and families, compassionate attitudes, and respectful communication, as well as the obvious need for adequate staffing and equipment. A summary of the comprehensive needs required to address this issue would be welcomed in the Discussion or Conclusion sections of this manuscript. If that is thought to be beyond the scope of this manuscript, I will look forward to a follow-up manuscript to address this important topic. Thank you for bringing focus to these issues that are of international significance, as I strongly suspect these problems are not limited to 5 hospital in Kenya.

Response to Reviewer 3: Thanks for your comments We have added in some potential solutions. However, since this part of the formative assessment to developing interventions, we would be describing the process of developing and implementing these interventions in subsequent follow up articles on what we were able to do as part of the larger study

---

## [Decision Letter · Decision Letter 1]

1 Nov 2021

PONE-D-20-28509R1Manifestations,  responses, and consequences of mistreatment of sick newborns and young infants and their parents in health facilities in KenyaPLOS ONE

Dear Dr. Abuya,

Thank you for submitting your manuscript to PLOS ONE. After careful consideration, we feel that it has merit but does not fully meet PLOS ONE’s publication criteria as it currently stands. Therefore, we invite you to submit a revised version of the manuscript that addresses the points raised during the review process.

The reviewers have suggested some minor additional revisions to your manuscript. Please assess and respond to these before a final decision can be made. Please submit your revised manuscript by Dec 16 2021 11:59PM. If you will need more time than this to complete your revisions, please reply to this message or contact the journal office at plosone@plos.org. Please include the following items when submitting your revised manuscript:A rebuttal letter that responds to each point raised by the academic editor and reviewer(s). You should upload this letter as a separate file labeled 'Response to Reviewers'.A marked-up copy of your manuscript that highlights changes made to the original version. You should upload this as a separate file labeled 'Revised Manuscript with Track Changes'.An unmarked version of your revised paper without tracked changes. You should upload this as a separate file labeled 'Manuscript'.If applicable, we recommend that you deposit your laboratory protocols in protocols.io to enhance the reproducibility of your results. Protocols.io assigns your protocol its own identifier (DOI) so that it can be cited independently in the future. For instructions see: https://journals.plos.org/plosone/s/submission-guidelines#loc-laboratory-protocols. Additionally, PLOS ONE offers an option for publishing peer-reviewed Lab Protocol articles, which describe protocols hosted on protocols.io. Read more information on sharing protocols at https://plos.org/protocols?utm_medium=editorial-email&utm_source=authorletters&utm_campaign=protocols.

We look forward to receiving your revised manuscript.

Kind regards,

Tanya Doherty, PhD

Academic Editor

PLOS ONE

Journal Requirements:

Reviewers' comments:

Reviewer's Responses to Questions

**Comments to the Author**

1. If the authors have adequately addressed your comments raised in a previous round of review and you feel that this manuscript is now acceptable for publication, you may indicate that here to bypass the “Comments to the Author” section, enter your conflict of interest statement in the “Confidential to Editor” section, and submit your "Accept" recommendation.

Reviewer #1: All comments have been addressed

Reviewer #2: (No Response)

2. Is the manuscript technically sound, and do the data support the conclusions?

Reviewer #1: Yes

Reviewer #2: No

3. Has the statistical analysis been performed appropriately and rigorously? 

Reviewer #1: N/A

Reviewer #2: No

4. Have the authors made all data underlying the findings in their manuscript fully available?

Reviewer #1: Yes

Reviewer #2: (No Response)

5. Is the manuscript presented in an intelligible fashion and written in standard English?

Reviewer #1: Yes

Reviewer #2: Yes

6. Review Comments to the Author

Reviewer #1: The paper is much clearer now and addresses better the complexity of this topic making the needed links with available literature. The authors may consider to present the conclusions in the same order of thinking as the categories of mistreatment, for example starting from the need of addressing health system challenges, including availability of competent neonatal health care providers, then the issues of building better communication and responding to the developmental needs of infants and emotional needs of families, and finally creating supporting environments for care and bereavement.

Reviewer #2: Abstract:

Conclusions need to reflect the results of the study. Summarize them in simple words.

Methods: I am sure this was not a cross sectional study. Please carefully write the methods section in the abstract as well as in the main methods section.

Findings of the study:

1. The analysis has been improved yet there needs to be effort for this analysis to meet the quality qualitative study expectations. A clear analytic approach is missing

2. The findings are merely description of what participants said and believed. It needs to go beyond and identify the nuances within the data and then add interpretation. Follow some quality qualitative articles on your topic.

3. The statement “An initial codebook of 19 themes and 100 sub-themes was developed and refined based on the process of open coding and progressive” is wrong. We can not have that many themes or subthemes in qualitative research.

4. Identify only a few themes by merging categories and tell a complete connected story about how your theme answer your research question. And what was your research question needs to be explicitly identified.

5. We describe what constitutes mistreatment of SYIs, what providers say drives their behaviors, parents’ immediate responses to mistreatment of SYIs, and the consequences of these experiences on parents: it is I think the parents would perceive and describe their experiences of mistreatment not you who will describe it. Therefore, your objectives need to be reset.

6. Where your are referring to table 1, I believe you mean Fig. 1.

7. Discussion will follow the amendments based upon the suggestions on revising the themes.

7. PLOS authors have the option to publish the peer review history of their article (what does this mean?). If published, this will include your full peer review and any attached files.

Reviewer #1: **Yes: **Dr Ornella Lincetto

Reviewer #2: **Yes: **Dr Jamil Ahmed

---

## [Author Response · Author response to Decision Letter 1]

15 Dec 2021

Comment from Reviewer #1: : The paper is much clearer now and addresses better the complexity of this topic making the needed links with available literature. The authors may consider presenting the conclusions in the same order of thinking as the categories of mistreatment, for example starting from the need of addressing health system challenges, including availability of competent neonatal health care providers, then the issues of building better communication and responding to the developmental needs of infants and emotional needs of families, and finally creating supporting environments for care and bereavement.

Response 1: We thank the reviewer for these comments. The conclusion has been revised to reflect the suggested changes and now reads as presented below

“This study outline types of mistreatment that were observed or reported for SYIs in five hospitals in Kenya and explores the responses and consequences of parents. Mistreatment for SYIs appears to be prevalent and linked to poor quality of care. To address mistreatment in this group of very young children, interventions that focus on building better communication, responding to the developmental needs of infants and emotional needs for the parents, strengthen the number of providers and their competencies in newborn care, as well as a supportive, enabling, and healthy environments, will lead to more respectful quality care for newborns and young infants” 

Comment from Reviewer #2: Abstract: Conclusions need to reflect the results of the study. Summarize them in simple words.

Response 2: This has been adjusted in line with review 1 and now reads

“Mistreatment for SYIs is linked to poor quality of care. To address mistreatment in SYI, interventions that focus on building better communication, responding to the developmental needs of infants and emotional needs for parents, strengthen providers competencies in newborn care, as well as a supportive, enabling environments, will lead to more respectful quality care for newborns and young infants”

Comment 3: Methods: I am sure this was not a cross sectional study. Please carefully write the methods section in the abstract as well as in the main methods section.

Response 3: This has been reworded in both the methods section of the study and the abstract. The corrected language in both sections removes the cross-sectional qualifier. It now reads

“This was a qualitative formative study designed as part of larger implementation research to develop and test strategies for improving provision of nurturing care; promotion of family engagement; and communication and respect for care of newborns, infants, and very young children in resource-constrained facilities in Kenya”

Comment 4-1 Findings of the study. The analysis has been improved yet there needs to be effort for this analysis to meet the quality qualitative study expectations. A clear analytic approach is missing

Response 4-1: The analytic approach – thematic analysis – is described on page 8 in the “data processing and analysis section.” We have added some language around our mixed approach (deductive and inductive) to guide the reader more carefully through our adaptation of existing frameworks and application to our data

Comment 4-2. The findings are merely description of what participants said and believed. It needs to go beyond and identify the nuances within the data and then add interpretation. Follow some quality qualitative articles on your topic.

Response 4-2. We appreciate this comment made in several rounds of review. However, we feel that the approach we have taken is analytical and integrates categories known from literature in the field and then we demonstrate manifestations using data as examples. The integrative approach we have used is deliberate to illuminate examples of typologies we present. This we hope provides evidence via the mundane details we present. 

Comment 4-3. The statement “An initial codebook of 19 themes and 100 sub-themes was developed and refined based on the process of open coding and progressive” is wrong. We cannot have that many themes or subthemes in qualitative research.

Response 4-3: We have removed the sub-theme categorization as it is irrelevant for this manuscript and retained the 19 themes which represent the main topic areas that contributed to the themes presented.

Comment 4-4. Identify only a few themes by merging categories and tell a complete connected story about how your theme answer your research question. And what was your research question needs to be explicitly identified.

Response 4-4: Our analysis was led both by deductive and inductive approaches and aligned to the existing typologies. Our story is then detailed using the typologies, but we go ahead and describe how these experiences affect caregivers of children and the reactions they exhibit. Our research intent was to describe perspectives of parents and providers, what constitutes mistreatment of SYIs, what drives provider behaviors, parents’ immediate responses to mistreatment of SYIs, and the consequences of these experiences on parents

Comment 4-5. We describe what constitutes mistreatment of SYIs, what providers say drives their behaviors, parents’ immediate responses to mistreatment of SYIs, and the consequences of these experiences on parents: it is I think the parents would perceive and describe their experiences of mistreatment not you who will describe it. Therefore, your objectives need to be reset.

Response 4-5: Appreciate the comment. We have reworded the objectives, by adding the clause “Drawing on perspectives of parents and providers,…we describe….” The use of “we” is an accepted way to draw the perspectives of the researchers into the interpretation of the data and study objectives – particularly relevant and helpful in qualitative work. What is meant here is, “The study describes…” We hope that with this change, the objectives are clearer.

Comment 6. Where you are referring to table 1, I believe you mean Fig. 1.

Response 6: This has been corrected 

Comment 7. Discussion will follow the amendments based upon the suggestions on revising the themes.

Response 7: We have responded to the issues on themes and how we have structured them. We believe the analytical approach resonates with literature. We therefore did not make much adjustment to the discussions except editorial changes.

We believe the changes made above are sufficient and adequately responds to the issues raised

We look froward for your final decision

Timothy Abuya, PhD

On behalf of the team

---

## [Editor Report · Decision Letter 2]

3 Jan 2022

Manifestations, responses, and consequences of mistreatment of sick newborns and young infants and their parents in health facilities in Kenya

PONE-D-20-28509R2

Dear Dr. Abuya,

We’re pleased to inform you that your manuscript has been judged scientifically suitable for publication and will be formally accepted for publication once it meets all outstanding technical requirements.

Kind regards,

Tanya Doherty, PhD

Academic Editor

PLOS ONE
---

## [Editor Report · Acceptance letter]

11 Feb 2022

PONE-D-20-28509R2 

  Manifestations, responses, and consequences of mistreatment of sick newborns and young infants and their parents in health facilities in Kenya 

Dear Dr. Abuya:

I'm pleased to inform you that your manuscript has been deemed suitable for publication in PLOS ONE. Congratulations! Your manuscript is now with our production department. 

Kind regards, 

on behalf of

Professor Tanya Doherty 

Academic Editor

PLOS ONE